# Deterministic and robust room-temperature exchange coupling in monodomain multiferroic BiFeO$_3$ heterostructures

W. Saenrang[1,2], B.A. Davidson[1,3,4], F. Maccherozzi[5], J.P. Podkaminer[1], J. Irwin[2], R.D. Johnson[6,7], J.W. Freeland[8], J. Íñiguez[9], J.L. Schad[1], K. Reierson[2], J.C. Frederick[1], C.A.F. Vaz[10], L. Howald[10], T.H. Kim[1], S. Ryu[1], M.v. Veenendaal[8,11], P.G. Radaelli[6], S.S. Dhesi[5], M.S. Rzchowski[2] & C.B. Eom[1]

Exploiting multiferroic BiFeO$_3$ thin films in spintronic devices requires deterministic and robust control of both internal magnetoelectric coupling in BiFeO$_3$, as well as exchange coupling of its antiferromagnetic order to a ferromagnetic overlayer. Previous reports utilized approaches based on multi-step ferroelectric switching with multiple ferroelectric domains. Because domain walls can be responsible for fatigue, contain localized charges intrinsically or via defects, and present problems for device reproducibility and scaling, an alternative approach using a monodomain magnetoelectric state with single-step switching is desirable. Here we demonstrate room temperature, deterministic and robust, exchange coupling between monodomain BiFeO$_3$ films and Co overlayer that is intrinsic (i.e., not dependent on domain walls). Direct coupling between BiFeO$_3$ antiferromagnetic order and Co magnetization is observed, with ~ 90° in-plane Co moment rotation upon single-step switching that is reproducible for hundreds of cycles. This has important consequences for practical, low power non-volatile magnetoelectric devices utilizing BiFeO$_3$.

[1] Department of Materials Science and Engineering, University of Wisconsin-Madison, Madison, WI 53706, USA. [2] Department of Physics, University of Wisconsin-Madison, Madison, WI 53706, USA. [3] CNR-Istituto Officina dei Materiali, TASC National Laboratory, Trieste I-34149, Italy. [4] Department of Physics, Temple University, Philadelphia, PA 19122, USA. [5] Diamond Light Source, Harwell Science and Innovation Campus, Didcot OX11 0DE, UK. [6] Department of Physics, University of Oxford, Oxford OX1 3PU, UK. [7] ISIS Facility, Rutherford Appleton Laboratory, Chilton, Didcot OX11 0QZ, UK. [8] Advanced Photon Source, Argonne National Laboratory, Argonne, IL 60439, USA. [9] Department of Materials Research and Technology, Luxembourg Institute of Science and Technology, Esch/Alzette L-4362, Luxembourg. [10] Swiss Light Source, Paul Scherrer Institut, 5232 Villigen PSI, Switzerland. [11] Department of Physics, Northern Illinois University, De Kalb, IL 60115, USA. Correspondence and requests for materials should be addressed to C.B.E. (email: eom@engr.wisc.edu)

Substantial effort has been devoted to understand the internal magnetoelectric coupling between ferroelectric (FE) and antiferromagnetic (AF) orders in $BiFeO_3$, motivated by interest in exchange coupling to a ferromagnetic (FM) overlayer for use in nonvolatile magnetoelectric devices[1–5]. Nearly all approaches to exchange coupling investigated so far rely on coupling between the magnetization of the ferromagnetic overlayer and the small canted moment in $BiFeO_3$ (~ 0.06 $\mu_B$ $Fe^{-1}$ [6]) that originates from the Dzyaloshinskii–Moriya interaction[7,8]. Due to the spin-cycloid structure found in single crystals of $BiFeO_3$, this net moment averages to zero over a cycloid period (~62 nm[3]) precluding coupling over macroscopic areas. Suppression of the spin cycloid (e.g., due to strain in thin films[9], the presence of domain walls in narrow stripes[10,11,12], or application of a large magnetic field[13,14]) can lead to the formation of a collinear $G$-type antiferromagnetic order with mutually orthogonal ferroelectric polarization vector (**P**), Néel vector (**L**), and canted moment (**M**$_c$) that allows the possibility of a macroscopic exchange coupling via **M**$_c$. While **P** is fixed along the pseudocubic (pc) < 111 > directions, in multidomain epitaxial $BiFeO_3$ films the orientation of **L** and **M**$_c$ can depend on the type (compressive, tensile) and magnitude of the strain (L. W. Martin, private communication;[15]).

To stabilize a one-to-one correlation between order parameters in $BiFeO_3$ and exploit this state for robust, deterministic exchange coupling without the complications of domain walls[16–20], we have fabricated Co/$BiFeO_3$ heterostructures with ferroelastic and ferroelectric monodomain properties in $BiFeO_3$ over the entire sample[21]. In bulk single crystals[22] and relaxed monodomain films[23], a nondeterministic exchange coupling mechanism between $BiFeO_3$ and Fe or Co overlayers has been shown to result from the energy degeneracy of the three allowed $\{112\}_{pc}$–type spin-cycloid planes. As a consequence, the first switching event in a single **P**-domain results in nucleation of multiple AF domains so that the original AF domain state cannot be recovered[24].

In the following we demonstrate that, unlike monodomain crystals and relaxed films, strained **P**-monodomain $BiFeO_3$ films are also AF-monodomain, characterized by a unique internal magnetoelectric state with a one-to-one relationship between **P** and the cycloid plane orientation that is reproducible after hundreds of switching events. This confirms that strain lifts the cycloid state degeneracy and modifies the internal magnetoelectric coupling of $BiFeO_3$. Consequently, we find that the magnetic order present at the interface with a ferromagnetic overlayer can be changed deterministically by polarization switching that is in turn correlated to the rotation of the magnetization of a Co overlayer. This robust exchange coupling mechanism offers an alternative solution for implementation in potential devices.

## Results

**Experimental overview.** To probe the interfacial magnetoelectric state of ferroelectric and ferroelastic monodomain $BiFeO_3$ films in the down ($r_1^-$) and up ($r_3^+$) polarization states (using the ferroelastic domain notation of ref. [25]), to correlate these to the rotation of Co ferromagnetic domains and understand the mechanisms of magnetoelectric coupling, a method for in situ ferroelectric switching during Photoemission Electron Microscope (PEEM) imaging was developed. Figure 1 shows a schematic of the experimental setup and summary of the ferroelectric–antiferromagnetic–ferromagnetic correlations we report here. PEEM imaging allows simultaneous spatially resolved monitoring of changes in ferromagnetic domains (by X-ray Magnetic Circular Dichroism, XMCD, on the Co $L_3$ edge) and AF domains in the interfacial $BiFeO_3$ (by X-ray Magnetic Linear Dichroism, XMLD, on the Fe $L_3$ edge), permitting unambiguous tracking of the spin orientations for the ferroelectric down (Fig. 1b) and up (Fig. 1c) states over many switching events. To complement the surface-sensitive information from PEEM imaging, the spin-cycloid properties in the bulk of the film were measured on separately prepared samples by large-area neutron diffraction (ND), also shown in Fig. 1. Magneto-optic Kerr effect (MOKE) measurements were performed at room temperature to confirm the exchange coupling between the $BiFeO_3$ and Co seen by XMCD-PEEM. These results together demonstrate a robust and deterministic exchange coupling between monodomain $BiFeO_3$ and Co at room temperature that can be readily exploited in practical devices.

**Antiferromagnetic analysis of the $BiFeO_3$ film.** The antiferromagnetic spin-structure in the bulk of the $BiFeO_3$ film and at the interface was studied by ND and XMLD-PEEM, respectively, the latter through light polarization angle scans whose electron-yield signal originates within <10 nm of the surface[26]. Analysis of ND results (Supplementary Note 2) in the down state (Fig. 2a, c) shows that the ferroelectric monodomain in $BiFeO_3$ possesses a single antiferromagnetic domain with the propagation vector in the film plane and a spin-cycloid plane orientation that is quite different from bulk single crystals, lying 12° from the film plane. XMLD-PEEM analysis shows that this cycloid continues unchanged up to the surface (Fig. 3a). This cycloid plane does not contain the polarization vector **P**, and is distinct from the three $\{112\}$–type planes seen in bulk single crystals[24] or the cycloid configuration observed previously in thin films[27], for which **P** lies in the cycloid plane. The cycloid propagation vector **k** extracted from the ND results is parallel to $[1\bar{1}0]$ (one of the allowed vectors in bulk, and along the substrate step direction) with a cycloid period of $66 \pm 2$ nm. The longer period as compared to that in bulk is in quantitative agreement with the value expected for a cycloid plane that does not contain **P**, as the period should scale inversely with the projection of **P** onto the plane (projection angle ~ 24°).

In the up state, ND results reveal a similar in-plane cycloid orientation (Fig. 2b, d) as seen in the down state, but XMLD-PEEM analysis (Supplementary Notes 6 and 7 and Supplementary Figs. 9 and 10) shows that a different antiferromagnetic order develops at the interface (Fig. 3b). These magnetoelectric configurations are maintained reproducibly after repeated cycling. The up state is characterized as a majority $r_3^+$ domain (71° switching) deduced by ND and piezoforce microscopy (Supplementary Note 1 and Supplementary Fig. 4); a minority domain (<12%) corresponding to 180° switching ($r_1^+$) is also found, which we disregard as its presence does not change the analysis or conclusions. ND identifies a single cycloid in the majority domain with orientation similar to that in the down state, but rotated 13° in the direction of the **P** rotation, with the same propagation vector. This direct experimental identification of a non-bulk-like spin-cycloid plane (including propagation vector and spin orientation) is consistent with evidence that strain can alter the cycloid properties in $BiFeO_3$, albeit differently than proposed in ref. [10] or reported in ref. [27]. The ND results demonstrate that the strain state of the film (Supplementary Fig. 1) creates a non-bulk, in-plane orientation of the cycloid only weakly dependent on the polarization vector.

In contrast, surface-sensitive XMLD polarization angular scans in the up state (Fig. 3b) are distinctly different from the down state, and are not consistent with the single-cycloid plane found by ND, nor a cycloid of any orientations found in bulk single-crystal $BiFeO_3$ (Fig. 3c, d and Supplementary Fig. 11). Instead, fitting the XMLD-PEEM results (Supplementary Note 7) requires

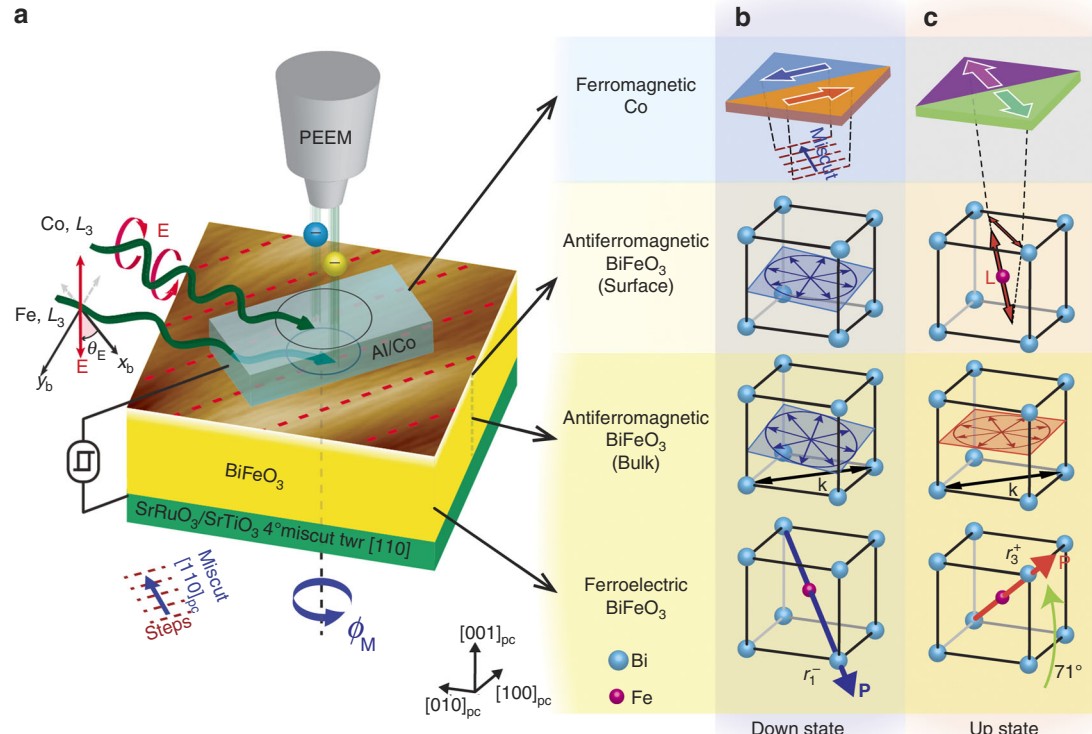

**Fig. 1** Experimental setup, monodomain BiFeO$_3$ and FE–AF–FM correlations. **a** Schematic diagram of in situ ferroelectric switching during PEEM imaging to study exchange coupling between monodomain BiFeO$_3$ and Co films. The magnetoelectric heterostructure is shown as mounted in the PEEM, with incoming circular and linear polarized light used for XMCD and XMLD measurements, respectively. The photon energy is tuned to the Co $L_3$ edge for the XMCD measurement and Fe $L_3$ edge for the XMLD measurement so that the ferromagnetic Co and antiferromagnetic BiFeO$_3$ layers can be selectively probed. The ferroelectric state of the BiFeO$_3$ can be switched in situ by 71° from the down state to the up state, allowing the properties of the ferromagnetic Co and surface antiferromagnetic BiFeO$_3$ layers to be characterized in the down and up states. Modeling of the surface antiferromagnetic spin-structure, whose results are shown in **b**, **c** ("BiFeO$_3$ (surface)"), is performed via polarization angle scans varying the linear polarization angle ($\theta_E$) at different sample azimuths ($\phi_M$). The antiferromagnetic state in the bulk of the BiFeO$_3$ film is additionally probed ex situ ND in both $r_1^-$ and $r_3^+$ states, also shown in **b** and **c**, respectively ("BiFeO$_3$ (bulk)"); a non-bulk-like spin-cycloid configuration of BiFeO$_3$ is observed in both down and up states. The surface AF order in the down state is cycloidal and identical to the cycloid seen by ND, while in the up state, a component of collinear AF order appears only at the surface with its axis in a vertical plane that includes the miscut direction. Correlated with the changes in surface antiferromagnetic order in BiFeO$_3$, the ferromagnetic moments of Co lie along the substrate steps in the down state and rotate ~ 90° toward the miscut direction in the up state. PEEM measurements were taken ~ 120 K. The BiFeO$_3$ (bulk) refers to properties averaged over the entire film thickness, for which contributions from near the interfaces are negligible

an additional component beyond the bulk cycloid measured by ND, either collinear antiferromagnetic order with **L** oriented along the [112]$_{pc}$ direction or a vertical cycloid plane with propagation vector along the substrate [110]$_{pc}$ miscut direction (discussed later). This extra component contributes ~ 25% of the total XMLD-PEEM intensity that should be distributed over 1 nm near the interface. The XMLD-PEEM data summarized here are identical for Co/BiFeO$_3$ and Pt/BiFeO$_3$ interfaces in both polarization states (within error bars).

**Ferromagnetic Co response upon BiFeO$_3$ polarization switching.** Demonstration of a one-to-one correlation between the magnetoelectric state of BiFeO$_3$ and Co magnetization is presented in Co XMCD-PEEM vector maps for a down-up-down switching sequence (Fig. 4) at 120 K. Analysis of the Co XMCD-PEEM images (Supplementary Note 8) yields vector magnetization maps of the Co magnetization showing a ~ 90° in-plane rotation upon ferroelectric switching. In the down state (Fig. 4a, c), the Co uniaxial anisotropy axis lies along the substrate step edges produced by the miscut (Supplementary Note 6 and Supplementary Figs. 2, 5); in the up state, the anisotropy axis rotates nearly 90° toward the [110]$_{pc}$ miscut direction (Fig. 4b). This correlation between Co anisotropy axis and BiFeO$_3$ polarization

demonstrates a previously unreported type of intrinsic exchange coupling between BiFeO$_3$ and Co that is deterministic, robust, and reliable over hundreds of switching cycles (Supplementary Note 5 and Supplementary Fig. 8). In the down state of BiFeO$_3$, Co domains are split parallel and antiparallel to the [1$\bar{1}$0]$_{pc}$ step edges direction, a distribution resulting from demagnetization effects that minimize the magnetostatic energy of the large electrode (100 × 100 μm²) and likely would not be present as the dimensions approach the micrometer scale[28]. The same anisotropy axis is seen in Co deposited directly on identically miscut (001) SrTiO$_3$ substrates (Supplementary Note 3 and Supplementary Figs. 5, 6). When the BiFeO$_3$ is switched to up state, the parallel/antiparallel Co domains rotate toward the [110]$_{pc}$ miscut direction (perpendicular to the step edges), as seen in the polar plots (Fig. 4d–f). Analyzing the local spin rotations, they are closer to ~ 75° upon switching (Supplementary Note 8 and Supplementary Fig. 12) leading to the overall distribution shown in the polar plots of Fig. 4d–f.

**Evidence of room temperature exchange coupling.** MOKE measurements at room temperature using the configuration shown schematically in Fig. 5a, b demonstrate the same exchange coupling seen in the XMCD-PEEM vector maps. **M**-**H** hysteresis

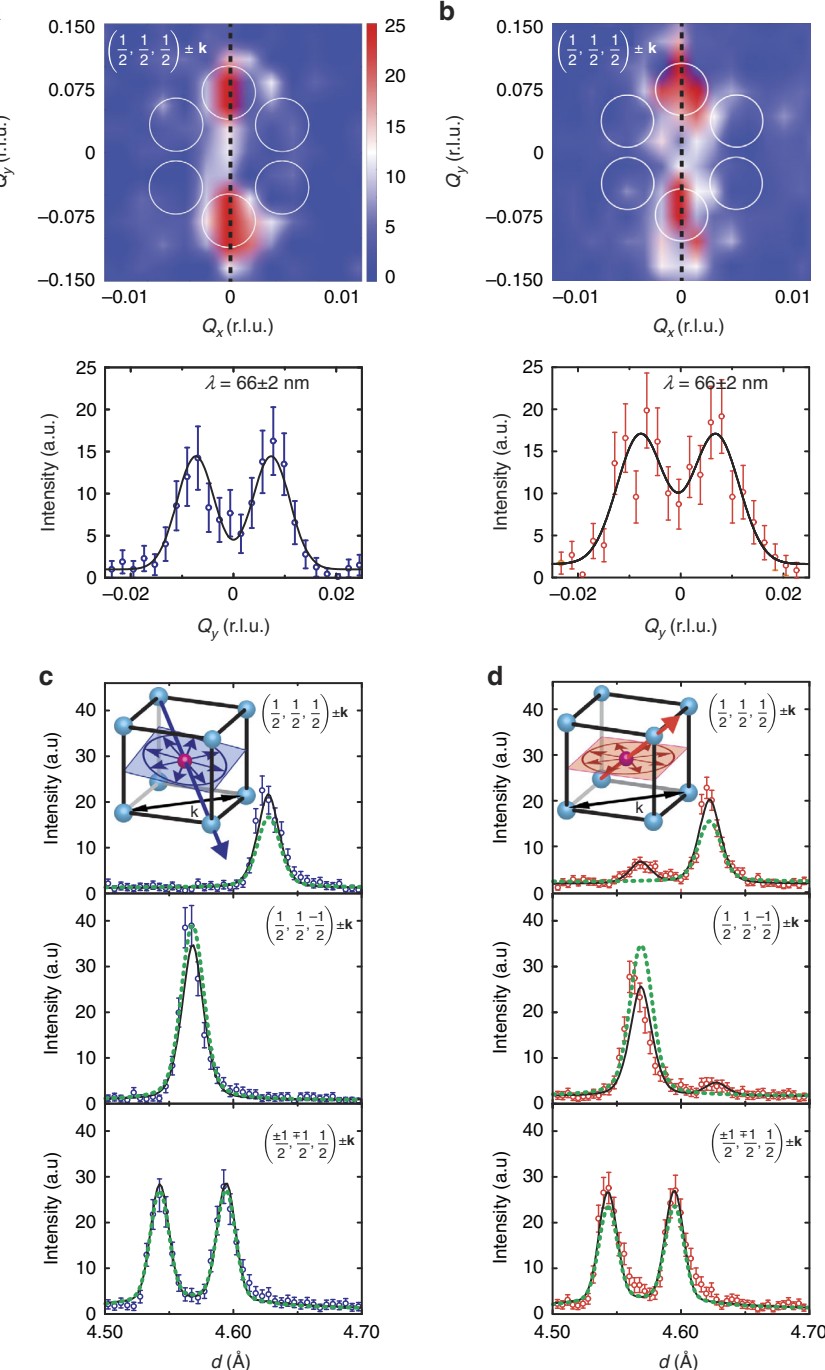

**Fig. 2** Distinct single-spin-cycloid configuration for each polarization state in the bulk of the BiFeO$_3$ film. **a, b** Shows reciprocal space maps (RSMs) of magnetic (½,½,½) ±**k** ND peaks (**k** is the cycloid propagation vector) measured in the down state and the up state, respectively. The positions of three, ±**k** pairs of peaks, corresponding to the three magnetic domains observed in bulk single crystals, are superimposed as white circles. Below RSMs, a 1D slice through $Q_x = 0$ (dotted black lines) from RSMs in which the peaks are fit (solid black lines) to determine the cycloid period λ. **c, d** Shows magnetic ND data measured from the down and up states, respectively. In the top and middle panels of both **c, d** the respective ±**k** satellites of the majority domain exactly overlap in d-spacing. In the panels of both **c, d** the ±**k** satellites are separated in d-spacing, as seen, and the symmetry equivalent (½,-½,½)$_{pc}$ ±**k** and (-½,½,½)$_{pc}$ ±**k** reflections are summed. Note that the crystallographic basis switches with the polarization such that, for example, the top panel of **c** and the middle panel of **d** are acquired with the same film orientation in the diffractometer. The fit model is shown as a black line, in which the lattice parameters, propagation vector, peak profile parameters, background, ferroelastic domain populations, and the tilt angle of the cycloidal plane were refined. ND identifies a single-cycloid plane lying 12° from in-plane in the down state that rotates to almost in-plane in the up state with the same sense of the polarization switching (inset of **c, d**). The dotted green lines are the magnetic ND intensity simulations of the bulk BiFeO$_3$ single-crystal spin-cycloid plane. The additional weak peaks in the panels for (½,½,½) ±**k** and (½,½,-½) ±**k** in **d** (up state) originate from a minority ferroelastic domain. The bulk of BiFeO$_3$ film refers to properties averaged over the entire film thickness, for which contributions from the near the interfaces are negligible. Error bars are given by ±$\sqrt{N}$, where N is the number of counts

loops (Fig. 5c, d) with the magnetic field applied along the $[110]_{pc}$ miscut direction and $[1\bar{1}0]_{pc}$ step edges direction show that the Co anisotropy axis rotates from parallel to the step edges in down state to nearly perpendicular in the up state. The hysteresis loops can be simulated using the Stoner–Wohlfarth model that includes two anisotropy energies, one from the steps present in both down

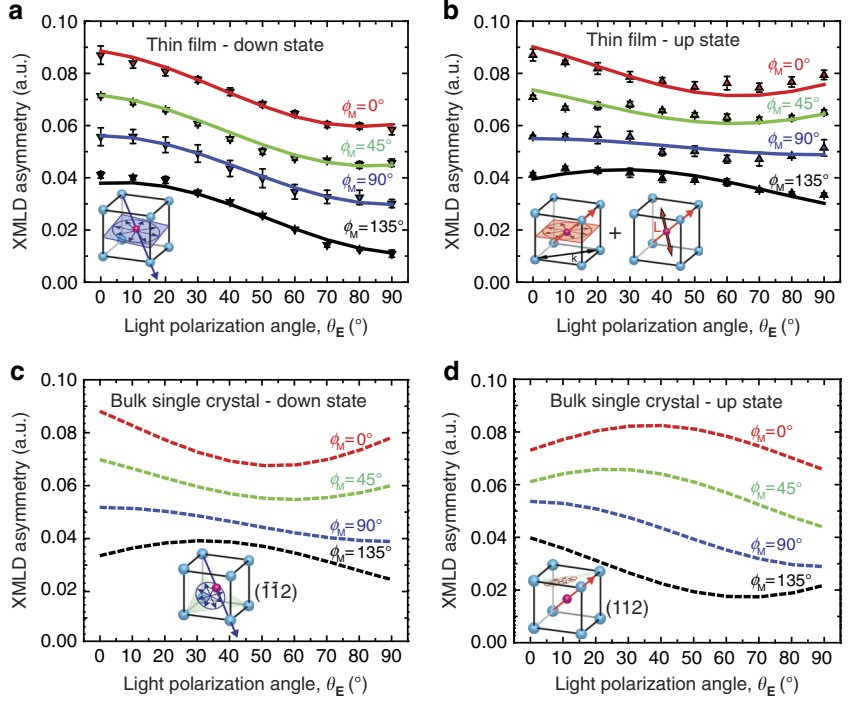

**Fig. 3** Surface antiferromagnetic order of BiFeO₃ films. **a**, **b** XMLD polar scans at the Fe $L_3$ edge of Pt(2 nm)/BiFeO₃ thin film for the down state and the up state, respectively. In each scan the light polarization angle $\theta_E$ varies from 0° to 90° for a fixed azimuthal angle $\phi_M$, where $\phi_M = 0°$ corresponds to the incident light wave vector parallel to the substrate steps $[1\bar{1}0]_{pc}$ direction. The continuous lines are best-fit results (Supplementary Note 6 and Supplementary Table 2) for the shown data points. For the down state **a**, fitting gives the same single-cycloid plane given by ND results within ~ 2°. For the up state **b**, the fitting assumes the same single-cycloid plane given by ND results with an additional surface collinear AF component with a weight of 25% (see text). Panels **c**, **d** are simulations for one of the bulk single-crystal ($\bar{1}\bar{1}2$) and (112) spin-cycloid planes (inset) in down state and up state, respectively. The error bars show the precision of the measurements defined as the standard deviation from the fits

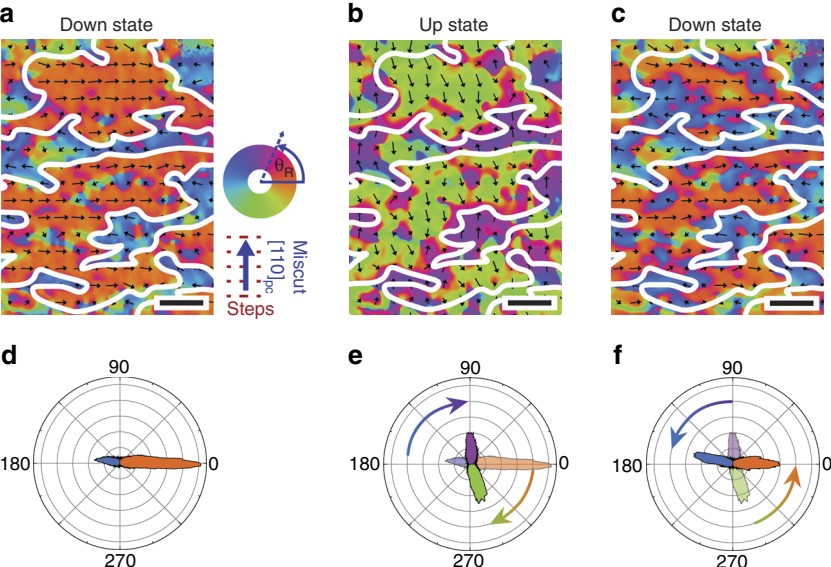

**Fig. 4** Magnetization rotation with ferroelectric switching. XMCD-PEEM vector maps of Co local magnetization for BiFeO₃ polarization switching from the **a** down to **b** up and back to **c** down state consecutively. Arrows in the vector maps give the local magnetization directions and the polar plots show the angular distribution of the Co magnetization. The domain boundaries are indicated by the white lines. Scale bar, 1 μm. The polar plots indicate an average Co magnetization rotation of nearly 90° from the **d** down state to **e** up state and then back to the **f** down state. The transparent parts in **e**, **f** are polar plots from **d**, **e**, respectively, to show the population distribution from previous state. No magnetic field is applied at any time during the measurements. The vector maps were taken at 120 K

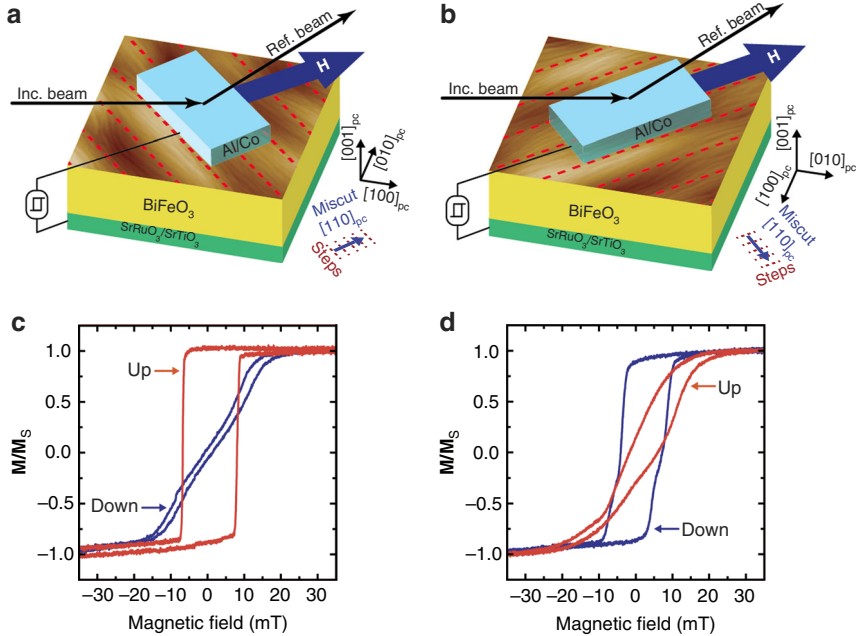

**Fig. 5** Room temperature exchange coupling measurement. **a** Schematic of the configuration used to measure MOKE **M-H** loops in the Co layer at room temperature with the magnetic field applied along the **a** $[110]_{pc}$ miscut direction and **b** $[1\bar{1}0]_{pc}$ step edges direction. The overlaid surface AFM image of BiFeO$_3$ film shows step edges (dotted red lines). **M-H** loops demonstrating an in-plane rotation of the Co easy magnetic axis upon ferroelectric switching of the BiFeO$_3$ with the magnetic field applied along the **c** $[110]_{pc}$ miscut direction and **d** $[1\bar{1}0]_{pc}$ step edges direction

and up states, and a second appearing in the up state that is oriented nearly along the miscut, as suggested by the XMLD-PEEM results shown in Fig. 3. In this model, fits to the hysteresis loops confirm the ~ 75° easy axis rotation upon switching seen in the XMCD vector maps (Supplementary Note 4 and Supplementary Fig. 7).

## Discussion

The combined experimental results suggest an exchange coupling mechanism across the Co/BiFeO$_3$ interface that controls the Co rotation upon switching. In the down state (equivalently, for Co deposited on miscut SrTiO$_3$) the magnetic easy axis is imposed by the miscut that breaks the symmetry between the otherwise equivalent in-plane directions. Since BiFeO$_3$ up and down states present exactly the same in-plane strain, no strain-mediated—piezomagnetic or magnetostrictive—coupling across the interface can explain the observed Co spin rotation. In contrast, up and down states present different polarization charge screening at the interface; in the up state an excess of metallic electrons accumulating at the interface is expected, while in the down state these will be replaced by positive carriers. Such changes in electronic density can potentially influence the behavior of both BiFeO$_3$ and Co close to the interface. In the case of ferromagnetic metals, such effects are known to affect the magnetic anisotropy between different crystallographic directions, and could potentially lead to a rotation of the easy axis[29]. However, experimentally we measure stronger coupling to Co for thinner BiFeO$_3$ films, while charge-driven effects should be independent of thickness; hence, we can tentatively disregard the direct effect of screening on the Co layer.

Finally, and unexpectedly, we have found a difference in the magnetic structure of BiFeO$_3$ near the interface for up and down states. Our ND data indicate that the two polarization states display a similar $(001)_{pc}$ cycloid plane in the bulk of the film, while XMLD analysis reveals that the up state presents an additional interfacial magnetic structure suggesting spins lying within the vertical plane containing the $[110]_{pc}$ miscut direction, most

likely collinear. This feature clearly breaks the symmetry between $[110]_{pc}$ miscut and $[1\bar{1}0]_{pc}$ step edges directions, and could be responsible for the preferred Co spin orientation in the up state. Two ingredients are needed for such a mechanism to be active, namely, exchange coupling between Fe and Co spins across the interface, and the presence of local ferromagnetic moments along the miscut direction in the interfacial BiFeO$_3$. While the former can be taken for granted, the latter is more delicate, but not implausible; indeed, such a situation may pertain to canted antiferromagnetic or cycloid structures, or if the near-interface spin-structure of BiFeO$_3$ were ferromagnetic or A-type anti-ferromagnetic (presenting ferromagnetically aligned spins within $(001)$-oriented FeO$_2$ planes). Our data are compatible with these possible configurations, but does not allow distinguishing between them.

Several concluding points are worth making regarding the unusual interfacial magnetic structure of BiFeO$_3$ in the up state driving the exchange coupling. First, epitaxial strain is known to affect the BiFeO$_3$ antiferromagnetic structure[10,27], with our results showing selection of a previously unseen cycloid variant in the bulk of the film and for the first time a polarization dependence in the surface order that correlates with the Co rotation. Second, accumulation of free carriers may favor specific magnetic interactions in the interfacial BiFeO$_3$, with the presence of extra electrons in the up state expected to favor ferromagnetic inter-actions that could lead, e.g., to sizeable localized canted moments[30]. Such a scenario is physically sound, and could explain the observed magnetoelectric control of Co spins. Finally, we note that the deterministic and robust exchange coupling reported here is a consequence of the magnetoelectric properties of monodomain BiFeO$_3$ films that has clear potential for applications: the observed in-plane Co moment rotation of ~ 90° over device-relevant areas is quite sufficient for large changes of tunneling magnetoresistance[31] in technologically useful magnetic tunnel junctions. The significance of these results lies also in the unique advancement of state-of-the-art approaches to multifunctional oxide film growth (domain engineering and

epitaxial strain in multiferroics) and characterization via spectroscopic and diffraction techniques available only at large-scale facilities (PEEM with in situ ferroelectric switching and ND) that are necessary to elucidate the microscopic characteristics of the magnetoelectric and exchange coupling.

## Methods

**Sample preparation and device fabrication.** (001) $SrTiO_3$ single-crystal substrates with 4° miscut toward $[110]_{pc}$ direction are used for domain engineering of $BiFeO_3$ thin films[21]. A 35 nm thick $SrRuO_3$ bottom electrode layer is first deposited by 90° off-axis sputtering[32] at 600 °C followed by 300 nm of $BiFeO_3$ films grown by double-gun off-axis sputtering at 750 °C with $Ar:O_2$ ratio of 4:1 at a total pressure of 400 mTorr[33]. The $BiFeO_3$ target contains 5% excess $Bi_2O_3$ to compensate for bismuth volatility[33]. Subsequently, 2 nm Co and 3 nm Al are deposited as the ferromagnetic and passivation layers by magnetron sputtering at room temperature without applied magnetic field. Capacitor structures were defined lithographically with a photoresist mask and subsequently ion-milled down to the $BiFeO_3$ film. The devices were wirebonded to a specially designed printed-circuit board and/or chip carrier for in situ ferroelectric switching measurements in the PEEM. The substrate miscut induces the growth of a single ferroelastic domain variant, while the $SrRuO_3$ bottom electrode introduces an electrical boundary condition favoring the ferroelectric down $r_1^-$ state[21] since the depolarization field can be screened by free charge in the electrode during growth[34]. 71° switching leads to the monodomain up $r_3^+$ state. Multiple (separate) top electrodes were patterned on a $5 \times 10$ mm$^2$ sample for ND experiments, ensuring >98% of the film volume was switched and confirmed by PE measurement without preset loop.

**PEEM.** The PEEM results were recorded on beam line i06 (Diamond Light Source, UK), which is equipped with two Apple-II undulators delivering a high flux of X-rays on a 10 μm diameter spot with tunable polarization in the energy range 80–2100 eV. The light polarization can be left or right circular or linear (with variable orientation of $\mathbf{E}$ in the range $0° \leq \theta_E \leq 90°$). The beam is incident on the sample at an angle of 16° and $\theta_E = 0°$ corresponds to s-polarization. The PEEM is an Elmitec SPELEEM-III equipped with a manipulator stage with motorized x, y translation and manipulator azimuthal rotation $\phi_M$, a liquid nitrogen cryostat and a Multiferroic Tester II (Radiant Technologies, Inc., Albuquerque NM) to determine the FE state of the device in situ. $\phi_M$ could be changed over a 200° range. The probing depth of the PEEM technique in electron yield is estimated to be ~ 5 nm. The cobalt FM domain structure was measured by combining XMCD-PEEM images at the Co $L_3$ edge taken at $\phi_M = 90°$ and 0° to obtain the local magnetization in-plane vector[35]. These XMCD-PEEM vector maps of the Co FM domain structure were performed after poling the sample either up or down with the FE tester. The FE state of the $BiFeO_3$ was checked with the FE Tester before and after each of the XMLD polar scans as shown in Supplementary Fig. 3.

**Neutron diffraction.** Single-crystal neutron diffraction experiments were performed using WISH, a time-of-flight diffractometer at ISIS, the UK Neutron and Muon Spallation source (Supplementary Note 2).

**Data availability.** The data that support the findings of this study are available from the corresponding authors on reasonable request.

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

## Acknowledgements

This work was supported by the Army Research Office through grant W911NF-10-1-0362 and W911NF-13-1-0486. PEEM measurements at Diamond Synchrotron (Didcot, UK) were performed at the i06 beamline under proposals nt12084, nt13225, and si11589. Part of this work was performed at the Surface/Interface: Microscopy (SIM) beamline of the Swiss Light Source, Paul Scherrer Institut, Switzerland. Work at Argonne National Laboratory was supported by the US DOE, Office of Science, Office of Basic Energy Sciences, under Contract No. DEAC02-06CH11357. M.v.v. is supported by the U.S. DOE under Award No. DE-FG02-03ER46097. R.D.J. acknowledges STFC for the provision of beam time on the WISH instrument at ISIS, proposal number RB1600019, and P. Manuel for data collection. R.D.J. acknowledges support from a Royal Society University Research Fellowship. J.Í. acknowledges support from the Luxembourg National Research Fund (Grant number FNR/P12/4853155).

## Author contributions

C.B.E., B.A.D., and W.S. conceived the project. W.S. grew and patterned the
$BiFeO_3$ heterostructures and performed AFM, PE, and PFM measurements. F.M., J.P.P.,
W.S., B.A.D., J.I., C.A.F.V., S.S.D., and L.H. took the PEEM data, and F.M. and
S.S.D. analyzed it. J.W.F., B.A.D., J.P.P., W.S., J.C.F., and S.R. took and analyzed the
XAS measurements. Neutron diffraction data were taken by R.D.J. and analyzed by
R.D.J. and P.G.R., and J.I., K.R., and W.S. took and analyzed the MOKE measurements.
M.v.V. and J.W.F. performed multiplet modeling of XAS/XMLD data. T.K. and
W.S. performed and analyzed XRD. J.Í. provided theoretical analysis. All authors
participated in discussion of the results and the manuscript. C.B.E. directed the
research.

## Additional information

**Competing interests:** The authors declare no competing financial interests.

**Reprints and permission** information is available online at http://npg.nature.com/
reprintsandpermissions/

**Publisher's note:** Springer Nature remains neutral with regard to jurisdictional claims in
published maps and institutional affiliations.

