## [Peer Review File · Nature Communications]

Reviewers' comments:

Reviewer #1 (Remarks to the Author):

This manuscript reports a carefully designed investigation on the magnetoelectric coupling in monodomain multiferroic BiFeO₃/Co heterostructures where the BiFeO₃ layer is deposited on properly mis-cut SrTiO₃ substrate so that the domain structure can be controlled to be monodomain. It is claimed that such a heterostructure allows the 90-degree in-plane magnetic domain switching upon the up-down polarization switching in the BiFeO₃ layer. This switching behavior is robust and well reproducible. Certainly, the exchange coupling in monodomain BiFeO₃ heterostructures is different from previous reports based on multistep ferroelectric switching with multiple domain structures.

The experimental data are interesting and the presentation is comprehensive. The referee thinks this paper may be acceptable in Nature Comm if the following issues can be properly addressed. Also, adding some of the experimental details can improve the readability of this manuscript.

The central issue I am concerned with is that the ferroelectric polarization in BiFeO₃ has two components, and the magnetically induced component is weak compared to the component induced by the Bi-6s lone pair mechanism. When the 70-degree polarization up-down switching occurs, how to clarify the role of the component associated with the spin-cycloid structure and the other one? and how about the coupling between the two components and the Co domains? In fact, the ND data in this work did reveal the difference between the spin-cycloid plane and the polarization in the “down” state, and I guess that the origin for this difference is the dominant polarization component from the Bi-6s lone pair over the magnetically induced component which is on the spin-cycloid plane.

There are also several points which may not be all minor:

- 1, The ferroelectric domain structures were not clearly shown in the submitted manuscript. I think the authors should show the vector PFM images (out-of-plane and in-plane) of BFO initial state, “down” state and “up” state to demonstrate the initial monodomain and the following 71° switch state after applying an external electric field. The quality of Fig. S4 should be improved too.

- 2, The authors use ferroelectric hysteresis loops before and after XMLD and XMCD PEEM

imaging to prove the polarization state stability. I don't know why the authors choose this method? Why the coercive force after PEEM measurements is larger than the initial state?

3, The authors report the full BFO heterostructures have a step morphology. I understand the step morphology reflected in Fig. S2 and Fig. S4. However, I think the authors need to improve the image resolution ratio to show the step morphology more clearly.

4, As mentioned in Fig. S5, BFO "up" and "down" states present exactly the same in-plane strain, no strain-mediated – piezomagnetic or magnetostrictive – coupling across the interface can explain the observed Co spin rotation. I wonder, if there any strain coupling across the interface during the switch process, and does this interfacial strain have impact on the Co spin rotation?

5, For the interfacial magnetoelectric coupling, especially the exchange coupling across the interface, via the DM mechanism induced spin canting, the authors may need to cite those recent works on this issue, e.g. PRL 103, 127201 (2009).

Reviewer #2 (Remarks to the Author):

The manuscript reports on a novel manifestation of magnetoelectric switching in a Co/BiFeO₃/SrRuO₃/SrTiO₃(001) heterostructure. The BFO film is 150 nm / 300 nm thick and grown on a miscut-controlled substrate that allows for growth of a single-domain BFO film and vertical switching of ferroelectric polarization. The Co in-plane magnetic easy axis can be rotated by about 90 degrees via switching the BFO polarization (Fig.5). On the nanoscale explored by XMCD-PEEM, most Co domains switch magnetization deterministically by 90 degrees in one direction (Fig.4). From neutron diffraction data, the BFO magnetic single-domain state for both orientations of polarization is derived as a spin spiral state (that deviates from that known for strain-free BFO single crystals). The BFO magnetic order near the Co/BFO interface is probed by XMLD in both polarization orientations. Interface-near BFO is found to show different magnetic order than the bulk magnetic spiral determined from neutron diffraction and strongly changes upon polarization reversal. The polarization "up" state seems to induce a collinear antiferromagnetic moment which is suggested as decisive ingredient for the Co coupling and switching.

These experimental findings may provide new insights into single-domain, long-life-time magnetoelectric switching of spintronic devices using BFO underlayers at room temperature, with strong practical implications. Further, they are of fundamental importance with regard of

understanding the magnetism of BFO interfaces. However, some of the presented results are not convincing enough and need to be clarified as pointed out below.

Twin-domain two-step electrical switching had been demonstrated at room temperature in 2014 for a spin valve of CoFe/Cu/CoFe on BFO (Ref.2), the major limitation in that stage being the small lifetime during cycled switching. As pointed out by authors of the present manuscript, removal of domain walls may be an important step to improve the lifetime. Experimental evidence for that in the present work is not clear to me. It mentions “reproducibility for hundreds of cycles” (abstract), but this is not shown in more detail, for instance, for hysteresis loops like those in Fig.5.

A large sample (BFO thickness, Co electrode area) is investigated because it is required for neutron diffraction. As a consequence, the term “monodomain” is not strictly true for the investigated sample, but it indicates it is likely to work with smaller samples. This should be clarified early in the text. A related point to be addressed is the undefined strain state of the thick BFO film; an evaluation about how this range of strains may affect magnetic order in BFO is required.

In Fig.2c, the bulk single crystal “down” state is surprisingly similar to Fig.2b, the BFO interface “up” state. Is this a coincidence?

Switching of Co is not reversible in some areas of Fig.4, where the rotation sequence is +90 degree and another +90 degree, ending in reversed state to the initial state. What is the reason for that, and how could it be improved?

The relative orientation of Co and Fe (staggered antiferromagnetic, L) moments is different for the Co domains shown in Fig.4, assuming a BFO single-domain state. What (magnetic coupling) mechanism would drive deterministic switching by 90 degrees for these different configurations?

Reviewer #3 (Remarks to the Author):

Saenrang et al. report on exchange bias coupling of single domain BFO and magnetic metallic layers at room temperature. While similar experiments have been described before for BFO, the novelty of having a single well defined domain state that has been extensively studied here can warrant publication in Nature Communications, in my opinion.

The study is well conducted and results reported are mostly discussed appropriately.

A few things need to be clarified:

- What is the exact direction of the cycloid in the up and down P state of the BFO thin films used for the study?
- What do the red dashed lines in Figs. 1 and 5 represent? Please state this clearly in the figure captions.
- Crystallographic directions in Fig. 5 could be added.
- Fig. S2 is missing scale bars.

Response on referees' comments

We would like to thank you and the reviewers for their in-depth reviews and excellent questions/suggestions regarding our manuscript. In the pages that follow, we provide our responses to each of their questions, in order. The responses are written in blue, and changes in a revised manuscript are marked in purple.

Reviewer #1 (Remarks to the Author):

This manuscript reports a carefully designed investigation on the magnetoelectric coupling in monodomain multiferroic BiFeO₃/Co heterostructures where the BiFeO₃ layer is deposited on properly miscut SrTiO₃ substrate so that the domain structure can be controlled to be monodomain. It is claimed that such a heterostructure allows the 90-degree in-plane magnetic domain switching upon the up-down polarization switching in the BiFeO₃ layer. This switching behavior is robust and well reproducible. Certainly, the exchange coupling in monodomain BiFeO₃ heterostructures is different from previous reports based on multistep ferroelectric switching with multiple domain structures.

The experimental data are interesting and the presentation is comprehensive. The referee thinks this paper may be acceptable in Nature Comm if the following issues can be properly addressed. Also, adding some of the experimental details can improve the readability of this manuscript.

The central issue I am concerned with is that the ferroelectric polarization in BiFeO₃ has two components, and the magnetically induced component is weak compared to the component induced by the Bi-6s lone pair mechanism. When the 70-degree polarization up-down switching occurs, how to clarify the role of the component associated with the spin-cycloid structure and the other one? and how about the coupling between the two components and the Co domains? In fact, the ND data in this work did reveal the difference between the spin-cycloid plane and the polarization in the “down” state, and I guess that the origin for this difference is the dominant polarization component from the Bi-6s lone pair over the magnetically induced component which is on the spin-cycloid plane.

We thank the reviewer for supportive and constructive comments. As the reviewer points out, the ferroelectricity in BFO is complex. Our main point is that we have experimentally identified differences in the spin cycloid configuration for up and down electric polarization, and correlated this with changes in the Co magnetic properties. The magnetic and lone-pair components of the polarization cannot be measured independently, but one may speculate that the magnetically induced polarization is oriented in the cycloidal plane, as is the case in the bulk. This would result in a different ratio between in-plane and out-of-plane components of the magnetic and lone-pair polarizations. As is clear in the conclusion section of the manuscript, we do not believe that the Co overlayer magnetization is directly coupled to either component of the polarization.

There are also several points which may not be all minor:

1, The ferroelectric domain structures were not clearly shown in the submitted manuscript. I think the authors should show the vector PFM images (out-of-plane and in-plane) of BFO initial state, “down” state and “up” state to demonstrate the initial monodomain and the following 71° switch state after applying an external electric field. The quality of Fig. S4 should be improved too.

We have reported extensively on the ferroelectric domain structure of these monodomain films in Ref. 22. Briefly, these results are summarized in Figs. S1, S2 and S3 from the Supplementary Information of Ref. 22. The XRD reciprocal space maps (RSMs) in Fig. S1 show that growth on STO with 4° miscut towards [110]_{pc} selects the r_1 variant over r_2 , r_3 and r_4 . The vector PFM imaging in Fig. S2 shows that screening of the BFO polarization by the bottom SRO electrode selects r_1^- over r_1^+ during growth; the selection as grown is 100% effective for BFO thickness of 300 nm. The XRD reciprocal space maps Fig. S3 shows that the stable “up” polarization state is r_3^+ for large enough ($> 3 \times 3 \mu\text{m}^2$) electrode areas, indicating single-step 71° switching upon polarization reversal.

To clarify the agreement between the data presented in the current manuscript with that of Ref. 22, we have expanded Supplementary Note 4 and Fig. S4 according to referee’s suggestion to include the vector PFM images. The 12x12 μm^2 scan area showing the ferroelectric domains is the largest that the PFM can scan, and these images confirm the BiFeO₃ films have uniform domains over large areas, with lower image resolution as a consequence (see response to comment 3 below).

2, The authors use ferroelectric hysteresis loops before and after XMLD and XMCD PEEM imaging to prove the polarization state stability. I don’t know why the authors choose this method? Why the coercive force after PEEM measurements is larger than the initial state?

A hysteresis loop is the easiest way to determine if the sample has retained its polarization, by the amount of polarization change on reversal. If the polarization had decayed, the full polarization value would not show up on reversal. The referee correctly points out that there is a 20% change in the coercive field. We do not have an explanation for this interesting observation, which, however, does not affect the main conclusions of the paper.

3, The authors report the full BFO heterostructures have a step morphology. I understand the step morphology reflected in Fig. S2 and Fig. S4. However, I think the authors need to improve the image resolution ratio to show the step morphology more clearly.

It is very difficult to resolve step structures for thin films on high miscut substrates because the terrace length is very small. We deliberately used high miscut (4 degree) to stabilize monodomain properties. Although it is hard to resolve the clear step morphologies, the steps are observable as small striations in Fig S2. The scale of Fig S4 is too large to clearly show the step morphology. Since Fig. S4 focuses on the large size of the ferroelectric domains in BFO films, the morphology images in Fig. S4 were removed to avoid confusion due to lower resolution of the images. (See also the response to comment 1.)

4, As mentioned in Fig. S5, BFO “up” and “down” states present exactly the same in-plane strain, no strain-mediated – piezomagnetic or magnetostrictive – coupling across the interface can explain the observed Co spin rotation. I wonder, if there any strain coupling across the interface during the switch process, and does this interfacial strain have impact on the Co spin rotation?

We are not aware of such a mechanism, but it can presumably be thought of as a continuous distortion of the BFO structure during the reversal process. Since this is likely reversible, we would expect that the magnetization would return to its original state, just as the BFO crystal structure does. Experimental confirmation of this hypothesis would be quite difficult and to our knowledge has not been reported in the literature, and is beyond the scope of this manuscript.

5, For the interfacial magnetoelectric coupling, especially the exchange coupling across the interface, via the DM mechanism induced spin canting, the authors may need to cite those recent works on this issue, e.g. PRL 103, 127201 (2009).

We have included the suggested reference as (new) Ref. 8 of the manuscript.

Reviewer #2 (Remarks to the Author):

The manuscript reports on a novel manifestation of magnetoelectric switching in a Co/BiFeO₃/SrRuO₃/SrTiO₃(001) heterostructure. The BFO film is 150 nm / 300 nm thick and grown on a miscut-controlled substrate that allows for growth of a single-domain BFO film and vertical switching of ferroelectric polarization. The Co in-plane magnetic easy axis can be rotated by about 90 degrees via switching the BFO polarization (Fig.5). On the nanoscale explored by XMCD-PEEM, most Co domains switch magnetization deterministically by 90 degrees in one direction (Fig.4). From neutron diffraction data, the BFO magnetic single-domain state for both orientations of polarization is derived as a spin spiral state (that deviates from that known for strain-free BFO single crystals). The BFO magnetic order near the Co/BFO interface is probed by XMLD in both polarization orientations. Interface-near BFO is found to show different magnetic order than the bulk magnetic spiral determined from neutron diffraction and strongly changes upon polarization reversal. The polarization “up” state seems to induce a collinear antiferromagnetic moment which is suggested as decisive ingredient for the Co coupling and switching.

These experimental findings may provide new insights into single-domain, long-life-time magnetoelectric switching of spintronic devices using BFO underlayers at room temperature, with strong practical implications. Further, they are of fundamental importance with regard of understanding the magnetism of BFO interfaces. However, some of the presented results are not convincing enough and need to be clarified as pointed out below.

Twin-domain two-step electrical switching had been demonstrated at room temperature in 2014 for a spin valve of CoFe/Cu/CoFe on BFO (Ref.2), the major limitation in that stage being the small lifetime during cycled switching. As pointed out by authors of the present manuscript,

removal of domain walls may be an important step to improve the lifetime. Experimental evidence for that in the present work is not clear to me. It mentions “reproducibility for hundreds of cycles” (abstract), but this is not shown in more detail, for instance, for hysteresis loops like those in Fig.5.

We thank the reviewer for supportive and constructive comments. We have added a new section in the Supplement (“Supplementary Note 8. Reproducibility and Robustness to Ferroelectric Cycling”) to directly address the reviewer’s comment, showing nearly identical MOKE hysteresis loops after 10 and 100 cycles of the BFO polarization.

A large sample (BFO thickness, Co electrode area) is investigated because it is required for neutron diffraction. As a consequence, the term “monodomain” is not strictly true for the investigated sample, but it indicates it is likely to work with smaller samples. This should be clarified early in the text.

Our BFO thin films are truly monodomain over the entire sample in the down state, and approximately monodomain in the up state (as mentioned in the text). The term “monodomain” refers to a ferroelectric and ferroelastic monodomain. Furthermore, we show by neutron diffraction (Supplementary Note 5) that our BFO is an antiferromagnetic monodomain in the bulk of the film over the same areal dimensions. We have clarified these points in the main text (page 4, last paragraph of Introduction), and also in the neutron section of the Supplementary Information. Experimentally, separate samples were prepared for the neutron experiments leaving >98% of the 5x10 mm² substrate covered by the BFO film and switchable via multiple electrically-isolated electrodes separated by 10 μm gaps. The PEEM and MOKE samples were patterned with much smaller electrode areas (typically 100 x 500 μm² per device, as discussed in Supplementary Note 3). The electrode configuration for the neutron samples was necessary to switch the entire area of BFO film over the 5x10 mm² substrate, while much smaller electrode areas could be used for the PEEM experiments since the typical field of view (FOV) was 20 μm diameter.

A related point to be addressed is the undefined strain state of the thick BFO film; an evaluation about how this range of strains may affect magnetic order in BFO is required.

We have characterized the strain state of our BFO films by X-ray diffraction reciprocal space maps, discussed in Supplementary Note 1, showing that our 300 nm films are partially relaxed. We agree with the reviewer on the importance of the effect of strain on magnetic order in BFO films, but note that there is very little literature with which to compare our data since cycloid properties in strained monodomain BFO films have not been reported previously. Sando et al (Ref. 9) report Mossbauer and Raman data measured on 70 nm BFO films on different nominally nonmiscut substrates (close to fully-strained, and probably multidomain though the domain structure was not reported) and give “indications” of how strain affects magnetic order, supported by calculation. Our results on monodomain films presented here do not match their conclusions of a bulk-like cycloid with the polarization lying in the cycloid plane that should be present at the strain state of our films (see Fig. 2 of Ref. 9). Bertinshaw et al (Ref. 28) have also used neutron diffraction but applied to 100 nm BFO films grown on STO (110) substrates, for which the strain state is quite different than on STO (001). We believe application of neutron diffraction as done

here or in Ref. 28 could experimentally resolve such questions and would be worth more attention in the future.

In Fig.2c, the bulk single crystal “down” state is surprisingly similar to Fig.2b, the BFO interface “up” state. Is this a coincidence?

We interpret the reviewer’s comment to refer to Fig. 1c and 1b, since the neutron diffraction data of Fig. 2 contains negligible contributions from the interface regions (thus Fig. 2b shows the “up” state in the bulk of the film, not the interface). It is true that in the bulk of the film probed by neutrons (results shown in Fig. 2), there is only a $\sim 12^\circ$ difference in cycloid plane orientation between ‘up’ and ‘down’ states. In Fig. 1, the interface spin model comes only from the PEEM results of Fig. 3 and not the neutron results of Fig. 2. In Fig. 1, the interfacial spin configuration in ‘up’ is quite different from ‘down’ because of the contribution of a collinear-type ordering, as discussed in the text.

We also understand the reviewer is possibly referring to Fig. 3b and 3c, which shows data with the fitting of the thin film “up” state at the interface and a simulation of one of the bulk single crystals. At a glance, the referee correctly points out that the fitting (dotted lines) in Fig. 3b is similar to the simulations (solid lines) in Fig. 3c. However, there are differences in term of the amplitude. The amplitude of the thin film “up” state is smaller than that of the bulk single crystal. To have a well fitted curves for Fig. 3b, we would have to incorporate the ND results with an additional surface collinear AF component with a weight of 25%.

Switching of Co is not reversible in some areas of Fig.4, where the rotation sequence is +90 degree and another +90 degree, ending in reversed state to the initial state. What is the reason for that, and how could it be improved?

This is an excellent question. When switching from the ‘up’ state back to the ‘down’ state it is energetically equivalent for a spin in the Co to rotate either +90 or -90 because the magnetic anisotropy is uniaxial. Because of this degeneracy and also because the PEEM measurements require zero magnetic field, there is a domain structure in the Co. The behavior pointed out by the referee corresponds to a slight motion of the domain wall. One way to improve on this might be to increase the domain wall pinning in the magnetic layer. Another way would be to apply a small bias field at 45° along a [100] direction which would then favor one domain variant in the ‘down’ state and one in the ‘up’ state.

The relative orientation of Co and Fe (staggered antiferromagnetic, L) moments is different for the Co domains shown in Fig.4, assuming a BFO single-domain state. What (magnetic coupling) mechanism would drive deterministic switching by 90 degrees for these different configurations?

Our argument, based on the observation of different surface magnetic structures by PEEM in the ‘up’ and ‘down’ states, is that different interfacial magnetic structure in the up polarization state

breaks the symmetry between the [110] and [1-10] directions. This symmetry breaking is responsible for the 90° rotation, independent of the original Co spin direction (parallel or antiparallel). That is, in the up state, the interfacial coupling affects the Co-spin component along [110], not that along [-110], forcing a rotation towards it in both Co-spin domains. An alternative explanation that does not require symmetry breaking is that the Co magnetization is quadratically coupled to the staggered magnetization. In this case, a change in magnitude of the coupling (from smaller to greater than the step anisotropy) will produce the observed 90-degree rotation of the Co magnetization. These two mechanisms are not necessarily exclusive, i.e. both could be present.

Reviewer #3 (Remarks to the Author):

Saenrang et al. report on exchange bias coupling of single domain BFO and magnetic metallic layers at room temperature. While similar experiments have been described before for BFO, the novelty of having a single well defined domain state that has been extensively studied here can warrant publication in Nature Communications, in my opinion.

The study is well conducted and results reported are mostly discussed appropriately.

We thank the reviewer for supportive and constructive comments.

A few things need to be clarified:

- What is the exact direction of the cycloid in the up and down P state of the BFO thin films used for the study?

The cycloid properties in the bulk of the BFO film are determined by neutron diffraction refinement, and are shown schematically in Fig. 2c and 2d, including both the propagation vector k direction and spin orientation; the magnitude of the propagation vector is given in the text. The interfacial spin ordering is determined from analysis of the linear dichroism in the PEEM polarization angle scans shown in Fig. 3, and agrees with the bulk ordering only for the 'down' state, i.e. the cycloid that exists in the bulk of the film propagates intact up to the interface. It is only in the 'up' state that the linear dichroism shows a more complex structure than just a cycloid. The exact direction of the cycloid in the "up" and "down" states in the bulk of the film is stated in the Section, "Antiferromagnetic analysis of the BiFeO₃ film" (starting on page 5):

The cycloid propagation vector k extracted from the ND results is parallel to [1-10] (one of the allowed vectors in bulk, and along the substrate step direction).

The direction is the same in the 'up' film, as stated on page 6:

'...with the same propagation vector.'

- What do the red dashed lines in Figs. 1 and 5 represent? Please state this clearly in the figure captions.

The red dashed lines indicate the orientation of the step edges. We have added this to Fig. 1 (below sample schematic) and Fig. 4.

- Crystallographic directions in Fig. 5 could be added.
We have added BFO crystallographic directions to the figure.

- Fig. S2 is missing scale bars.
We have added the scale bars to Fig. S2.

ADDITIONAL CHANGES TO THE MANUSCRIPT NOT DESCRIBED ABOVE.

Following the manuscript checklist, we have:

- moved references from abstract to main text
- labeled sections (Abstract, Introduction, Results, Discussion) and subsections (Results: Experimental Overview, Antiferromagnetic analysis of the BiFeO₃ film, Ferromagnetic Co response upon the polarization switching of the BiFeO₃ film, Evidence of room temperature exchange coupling)

Reviewers' Comments:

Reviewer #1 (Remarks to the Author):

I believe that all of my concerns with this work have been properly addressed in the revised manuscript and I have no additional comments on the technical aspects.

Reviewer #2 (Remarks to the Author):

The revised manuscript by Saerang et al. has improved in clarity of the key results. I believe it is appropriate for publication in Nature Communications now. This work demonstrates robust room temperature switching of a ferromagnetic Co film by electrically switching the polarization of a coupled BiFeO₃ monodomain film, allowing reproducible cycling of more than 100 times with moderate fatigue. This is a crucial advance over earlier work (e. g., Ref.17) where reproducible cycling was hard to achieve. Switchable interfacial magnetic order deviating from that of the bulk of BiFeO₃, as an essential ingredient for the coupling to Co, is discovered by combining neutron and X-ray absorption experiments. The coupling mechanism is different from that in previously reported experiments and is not fully understood at present. The latter point is well addressed in the present text and will drive further research. The comprehensive experimental study provides a breakthrough in revealing fundamental features of magnetoelectric interfacial coupling as well as a pathway towards practical use of BiFeO₃ for electric control of magnetic devices. For these reasons and the high overall quality of the manuscript, I have been convinced to recommend publication.

Reviewer #3 (Remarks to the Author):

Previous comments have been satisfactorily addressed. In my opinion the manuscript is now in publishable form.